# Nitroalkanes as thioacyl equivalents to access thioamides and thiopeptides

Xiaonan Wang[1], Silong Xu[1], Yuhai Tang[1], Martin J. Lear [2], Wangxiao He [3] & Jing Li [1] ✉

Thioamides are an important, but a largely underexplored class of amide bioisostere in peptides. Replacement of oxoamide units with thioamides in peptide therapeutics is a valuable tactic to improve biological activity and resistance to enzymatic hydrolysis. This tactic, however, has been hampered by insufficient methods to introduce thioamide bonds into peptide or protein backbones in a site-specific and stereo-retentive fashion. In this work, we developed an efficient and mild thioacylation method to react nitroalkanes with amines directly in the presence of elemental sulfur and sodium sulfide to form a diverse range of thioamides in high yields. Notably, this convenient method can be employed for the controlled thioamide coupling of multifunctionalized peptides without epimerization of stereocenters, including the late stage thioacylation of advanced compounds of biological and medicinal interest. Experimental interrogation of postulated mechanisms currently supports the intermediacy of thioacyl species.

Peptides have emerged as promising drug modalities and offer a number of advantages over small molecule drugs, including simpler design, straightforward synthesis, decreased immunogenicity, enhanced tissue penetration, and the ability to interact with under-explored targets[1]. However, their relatively short plasma half-life and low uptake by cells often limit their direct translation to the clinic[2]. Chemical modifications are frequently required to improve the ability of peptides to resist enzymatic degradation and to increase the pharmacological or drug-like properties[3]. The thioamide is the closest isostere of an amide with the same number of atoms, bond planarity, 3D shape and similar electronic properties. Nevertheless, the single-atom substitution of O to S renders intriguing chemical properties of thioamides over those of oxoamides, including unique thermodynamics[4], enhanced conformational rigidity[4,5], improved proteolytic stability[6], and characteristic spectroscopic profiles[7]. Due to these subtle but profound changes, the thioamidation of biologically active peptides, although often difficult to realize synthetically, is highly desirable by medicinal chemists to improve the thermal/proteolytic stability and the pharmacokinetic properties of amide-containing drug candidates. For example, N-cyclohexylethyl-ETAV, a plasma-stable and cell-permeable inhibitor, with a site-incorporated thioamide in the backbone, gave a 28-fold extended metabolic period in human blood than its oxoamide counterpart (Fig. 1a)[8]. Also the thioamidation of phenylpropionamide of the cyclic peptide anticancer cilengitide gave remarkable stability to thioamidated macrocyclic peptides in human serum (half-life 2160 min.) as compared to the all-oxo variant (half-life 540 min.)[5]. In nature, thioamides are accessed via various biosynthesis pathways to produce mono- to multiply-substituted thioamidated peptides with diverse biological properties[9]. Saalfelduracin A, for example, is a 26-membered cyclo-peptide synthesized ribosomally that is post-translationally modified via Thz11-Thr12 as a thiopeptide by a drug-resistant gram-positive pathogenic bacterium, which profoundly increases its antibacterial activity (16 times higher than its oxoamide family member Saalfel-duracin D)[10].

Although plagued by poor regioselectivity, chemoselectivity and yield issues, thioamide-containing peptides are traditionally synthesized by the conversion of oxoamides into thioamides, often by replacing O by S with the use of noxious and odorous thionating agents (Fig. 1b)[11–13]. Alternatively, thioacylating fragments of the

[1]School of Chemistry, and Xi'an Key Laboratory of Sustainable Energy Materials Chemistry, Xi'an Jiaotong University, 710049 Xi'an, China. [2]School of Chemistry, University of Lincoln, Brayford Pool, Lincoln LN6 7TS, UK. [3]The First Affiliated Hospital of Xi'an Jiao Tong University, 710061 Xi'an, China. ✉e-mail: jingli@xjtu.edu.cn

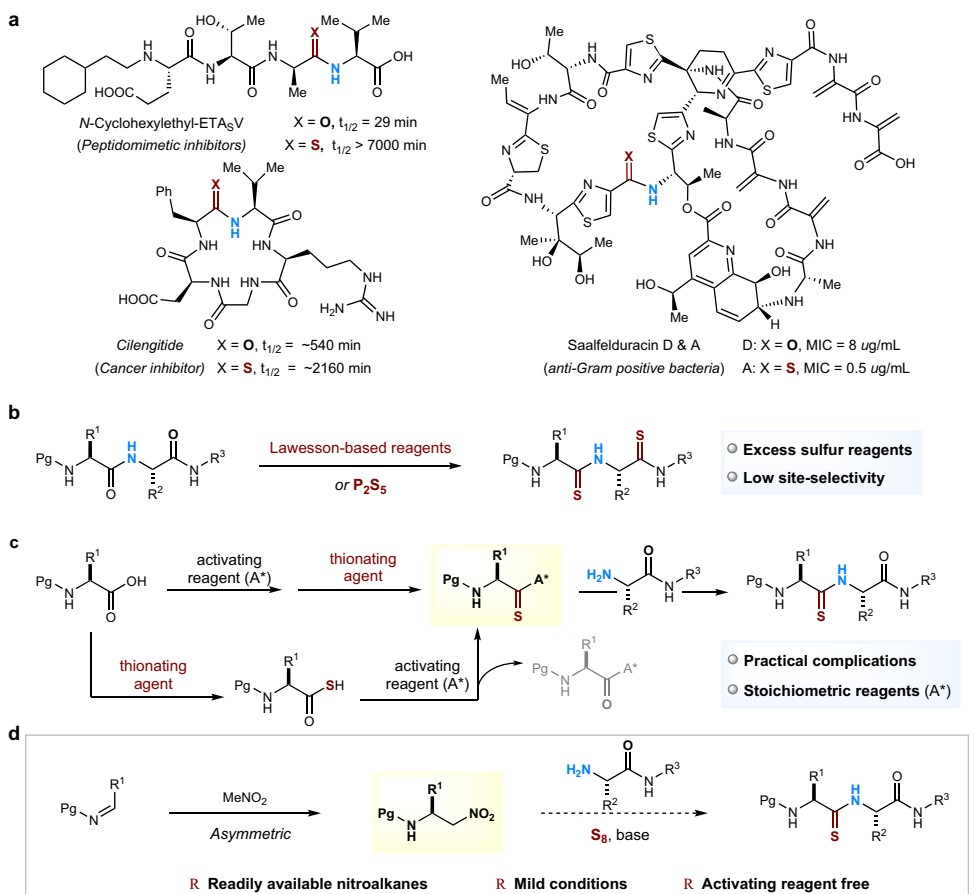

**Fig. 1 | Synthetic approaches to bioactive thioamides. a** Examples and importance of thioamide moieties in bioactive compounds. **b** and **c** Current strategies and common limitations for thioamide synthesis. **b** An O → S exchange approach requires the use of excess sulfur reagents. **c** A multi-step sulfur incorporation approach to making thioacylating substrates to generate thioamides. **d** *This work*: A direct and general method to thioacylate amines with nitroalkanes using elemental sulfur.

*C*-terminus of the peptide with preinstalled thiocarbonyl groups need to be prepared over multiple steps prior to the attachment of the *N*-terminus peptide amines (Fig. 1c). Such thioacylating fragments are either synthesized from the activation of acids, followed by replacing O by S with Lawesson-based thionating reagents (Fig. 1c, up)[14–18], or by the conversion of carboxylic acids into thioacids and then treatment with stoichiometric activating reagents (Fig. 1c, down)[19–21]. These strategies often resort to the use of excess activating reagents and suffer from practical difficulties on the bench, thereby limiting substrate scope and control.

Nitroalkanes of various complexities are readily available building blocks for synthesis. For instance, there is a wide variety of catalytic asymmetric methods for the direct preparation of α-chiral nitroalkanes from commercially available nitromethanes by reaction with imines, aldehydes, and so forth[22–24]. In particular, primary nitroalkanes are widely used to synthesize carboxylic acids, which was first reported by Kornblum in 1956[25,26] and later advanced by Mioskowski in 1997[27]. In 2010, Johnston and coworkers pioneered the concept of umpolung amide synthesis (UmAS) via the oxidative coupling of α-bromo nitroalkanes with amines in the presence of *N*-iodosuccinimide and potassium carbonate[28]. The Johnston group further expanded their UmAS concept to an extensive range of chiral nitroalkanes and amines[29–34]. Inspired by this body of work and our own amidation studies with the Hayashi group[35], we herein show the synergistic combination of readily available chiral nitroalkanes with amines and elemental sulfur provides a direct and efficient method to form thioamides and thiopeptides in excellent yields (Fig. 1d). This one-pot thioamidation method effectively solves the multiple selectivity

problems encountered when using Lawesson-type reagents and circumvents several practical complications in current synthetic sequences to activated thioacyl precursors.

## Results and discussion

### Reaction discovery and optimization

Our study began by exploring the reaction of the primary nitroalkane **1a** with 3-phenylpropanolamine **2a** in THF in the presence of $K_2CO_3$, where elementary $S_8$ was selected as the source of sulfur due to its low price, facile manipulation, and odorless nature (Table 1, entry 1). These conditions gave the desired thioamide **4a** in 37% isolated yield. Varying the base demonstrated $Na_2S$ (entry 4) to give superior yields of thioamide **3a** over inorganic bases such as $Na_2CO_3$, $Cs_2CO_3$ or $K_2CO_3$ (cf. entries 1–3 and entry 4). Soluble nitrogen bases such as DBU, $Et_3N$, pyridine was also found to be inferior to $Na_2S$ (cf. entry 4 and entries 5–7, Supplementary Table 1 for detailed optimization of solvent). No extra base gave thioamide **3a** in low yield (12%) due to the amine **2a** acting as a base, not as a reactant (entry 8). Although other electrophilic sulfur species were found to give comparably good yields (entries 9–12), elemental sulfur $S_8$ (2.0 equiv.) with $Na_2S$ (2.0 equiv.) stood out to afford **4a** in excellent yield and operational convenience (entry 4).

### Substrate scope

With optimized conditions identified, the substrate scope of various primary and secondary amines was investigated with the nitroalkane **1a** ($R^1 = CH_2CH_2Ph$; Fig. 2a). Primary amines carrying aliphatic, aromatic, alcohol, and alkene units were tolerated well under the reaction

**Table 1 | Optimization of thioamide synthesis by reacting nitroalkanes with various sulfur sources[a]**

| Entry | S source | Base | Yield (%)[b] |
|---|---|---|---|
| 1 | $S_8$ | $K_2CO_3$ | 37 |
| 2 | $S_8$ | $Na_2CO_3$ | 24 |
| 3 | $S_8$ | $Cs_2CO_3$ | 28 |
| 4 | $S_8$ | $Na_2S$ | 98 |
| 5 | $S_8$ | $Et_3N$ | 16 |
| 6 | $S_8$ | DBU | <5 |
| 7 | $S_8$ | pyridine | <5 |
| 8 | $S_8$ | no base | 12 |
| 9 | 3a | $Na_2S$ | 62 |
| 10 | 3b | $Na_2S$ | <5 |
| 11 | 3c | $Na_2S$ | 50 |
| 12 | 3d | $Na_2S$ | 70 |

THF tetrahydrofuran, DBU 1,8-Diazabicyclo[5.4.0]undec-7-ene.
[a]Reaction conditions: Nitroalkane 1a (0.2 mmol), amine 2a (0.4 mmol), S source (0.4 mmol), base (0.4 mmol) in THF (2 mL, 0.1 M).
[b]Isolated yield.

conditions, providing the thioamides 4a–4e in high yields. Notably, besides secondary amines reacting efficiently (4f–4h), the stereo-integrities of the α-positions of chiral primary amines were retained without racemization, giving the thioamides 4j–4l in good yields. When primary nitroalkanes bearing valuable functionalities such as –OH, –CO$_2$Me, –Cl, and acetal were used, the corresponding thioamides 5a–5m formed efficiently (Fig. 2b). Besides nitromethane giving quick access to thioformamide 5h, nitroalkyl compounds bearing α-functionality smoothly furnished thioamides 5i–5m. Subsequently, we applied this method to prepare thioamide pharmaceuticals and their analogues (Fig. 2c). Examples include a selective σ1 agonist[36] analogue 6a, a transcriptional antiestrogen[37] S-analogue 6b, a late-stage functionalized antidepressive drug, amoxapine 6c, a ciprofloxacin alogue 6d, which were all synthesized in good yields. Such late-stage functionalizations of either primary or secondary amine positions clearly adds value to medicinal or chemical biology pursuits and readily allows for the thioamide diversification of chemical libraries from advanced intermediates.

Having established the general synthetic utility of the thioamidation method, we set out to synthesize more complex thiopeptides using readily available α-chiral nitroalkanes (7)[38–41] and α-amino ester-terminated fragments. Under the basic conditions of Na$_2$S and S$_8$ at room temperature, the nitroalkane (R)-7a (ee > 99%) and 3-phenylpropylamine 2a formed the desired thioamide 8a in 96% yield and 76% ee, indicating partial epimerization over the reaction time (Fig. 3, entry 1, column 2). Further studies indicated that the reaction temperature and the sulfur reagent critically affected the epimerization and chemical yield during the thioamidation process (Fig. 3, see Supplementary Table 2 optimization of sulfur reagent and temperature). When elemental S$_8$ was used as the sulfur source, the enantioselectivity of 8a dramatically increased but the chemical yield decreased as the temperature was lowered. When S$_8$ was changed to the disulfide 3d, the enantioselectivity improved at lower

temperatures whilst keeping yields high. Optimal conditions were found over 36 h with the sulfur reagent 3d and Na$_2$S in THF at −10 °C (Fig. 3, entry 2, column 3). Here, enantiomerically pure nitroalkane (R)-7a (99% ee) and 3-phenylpropylamine 2a coupled to produce the desired thioamide 8a in 81% yield with no loss of stereogenic integrity.

With optimal conditions in hand for more complex substrates, the coupling of α-chiral nitroalkanes (R)-7 with a range of α-amino acid esters were screened (Fig. 4). This provided a significant level of generality and stereoselectivity without resultant epimerization of the thiopeptide products (10–11). Of note is the efficient coupling of sterically hindered secondary amines like proline (10u) and the use of unprotected side chains for serine, tyrosine, tryptophan, and histidine (10p/j/l/q). In addition, L-valine, L-Leucine, L-phenylalanine and the unnatural amino acid L-cyclohexyl glycine also couple readily with the L-tyrosine tert-butyl ester to afford thiopeptides 10v–10x in good yields. (Fig. 4a) Furthermore, the α-chiral nitroalkane (R)-7 (R$^1$ = CH$_2$CH$_2$Ph) reacted well with an amino dipeptide to produce the non-proteogenic tripeptide 11a in excellent diastereoselectively under these thioamidation conditions. Thioamide coupling between previously prepared tripeptide nitro-compounds and commercial amino acid esters also took place smoothly to afford the tetrapeptides 11b and 11c, bearing non-proteogenic residue groups, in moderate yields (Fig. 4b). In practice, 11b could also be prepared in similar yields in DMF or DMF/THF mixtures, which is especially useful for substrates that are poorly soluble in THF (Supplementary Table 3). In addition, a dipeptide nitroalkane was prepared, which not only coupled very well with the lysine amine of pal-tripeptide-1 to give 11d in 73% yield, but also with an NH$_2$-tetrapeptide to give the hexapeptide 11e in moderate yield. Unnatural α-aryl amino esters, although form thioamides under our current conditions, they partially racemize even at very low temperatures, as illustrated with (R)-phenyl glycine and amine 2a giving thioamide 8b. To further demonstrate the synthetic potential of this thioamidation method, a quick and convergent synthesis of

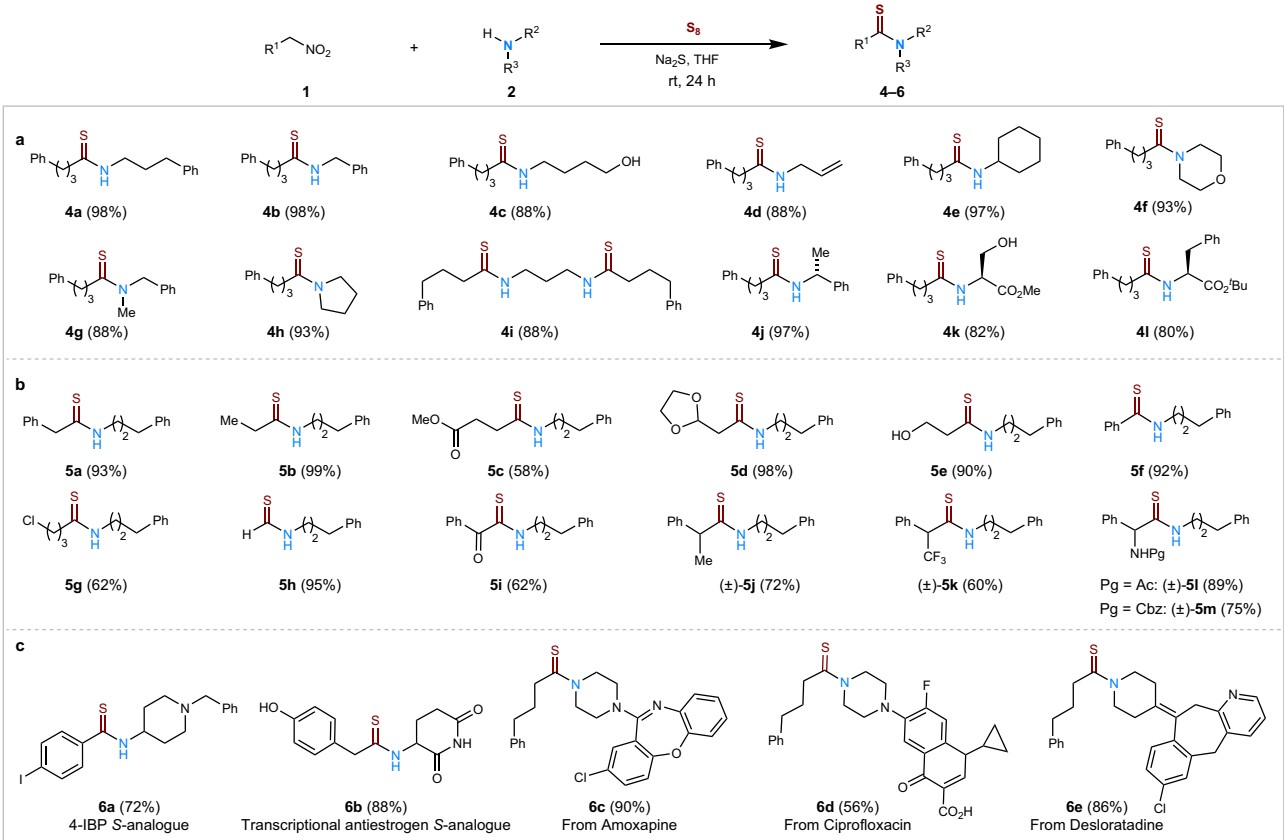

**Fig. 2 | Substrate scope and synthetic utility of the thioacylation of amines with nitroalkanes.** Reaction conditions: Nitroalkane **1** (0.2 mmol), amine **2** (0.4 mmol), S$_8$ (0.4 mmol), Na$_2$S (0.4 mmol), THF (2 mL), 24 h, rt; **a** Use of primary, secondary, and α-chiral amines. **b** Use of multifunctionalized nitroalkanes. **c** Late-stage thioacylation of bioactive substrates.

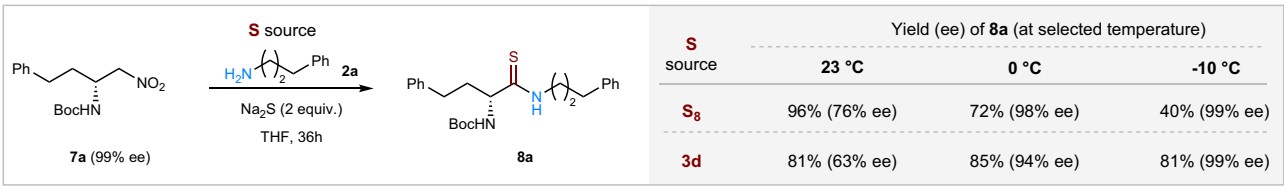

**Fig. 3 | Effect of temperature and sulfur (S) source on the stereogenicity of 8a.** Reactions carried out with 0.1 mmol of **7a**, 0.2 mmol of **2a**, 0.2 mmol of S source and 0.2 mmol of Na$_2$S in 1 mL of THF. Yield of isolated product. The ee value was determined by chiral-phase HPLC.

closthioamide, a highly potent natural antibiotic featuring six thioamide bonds[42], was achieved (Fig. 4c). The synthesis was accomplished in a straightforward manner from nitroalkanes **12** and **15** and propane-1,3-diamine by iterative dual thioamidations over 6 steps in 30% overall yield. Taken together, this method to form thioamide bonds provides a conceptually and reliable way to construct diverse thiocarbonyl polypeptides with no loss of stereogenic integrity and strategically allows ready preparation of mixed oxo/thioamide congeners of bioactive natural product and medicinal leads.

## Mechanistic studies

We next turned our attention to shedding light on the reaction mechanism (Fig. 5). Based on our observations and literature reports[43–47], there are three plausible mechanistic pathways to form thioamides from primary nitroalkanes and amines in the presence of electrophilic sulfur sources. One possibility is that the primary nitroalkane reacts with the amine and generates an oxoamide **17** first[31,35], which is then converted into its thioamide **4a** by the sulfur reagents (Fig. 5a, Path A). However, when the pre-made amide **17a** was

exposed to our standard reaction conditions, no thioamide **4a** was observed and only the oxoamide **17a** was recovered (Fig. 5b). Also, when pre-formed amide **17a** and primary nitroalkane **1a** and 1-butylamine were exposed to our standard reaction conditions, only the amide **17a** and the thioamide product **4l** were identified and isolated (no mixed thioamides were detected). These reactions clearly rule out oxoamide formation prior to thioamide formation. Another possibility is that the amine reacts with electrophilic source of sulfur and generates an electrophilic amine species such as **18** (Fig. 5a, Path B)[28]. This electrophile then reacts with nitroalkane **1** in the presence of base to generate an α-amino nitroalkane **19**, which further reacts with sulfur species to afford thioamide **4**. To test this possibility, the sulfur amine **18a** was premade and treated with the sodium salt of **1b**, but no reaction was observed (Fig. 5c). In addition, the proposed α-amino nitroalkane **19a** was pre-made via a reported procedure[44] and was treated to the thioamidation conditions and no thioamide **21** was detected, only the starting nitroalkane **19a** was recovered. As a third possibility, the nitroalkane reacts with the sulfur reagents and generates a thiocarbonyl intermediate **20** in situ, which is subsequently

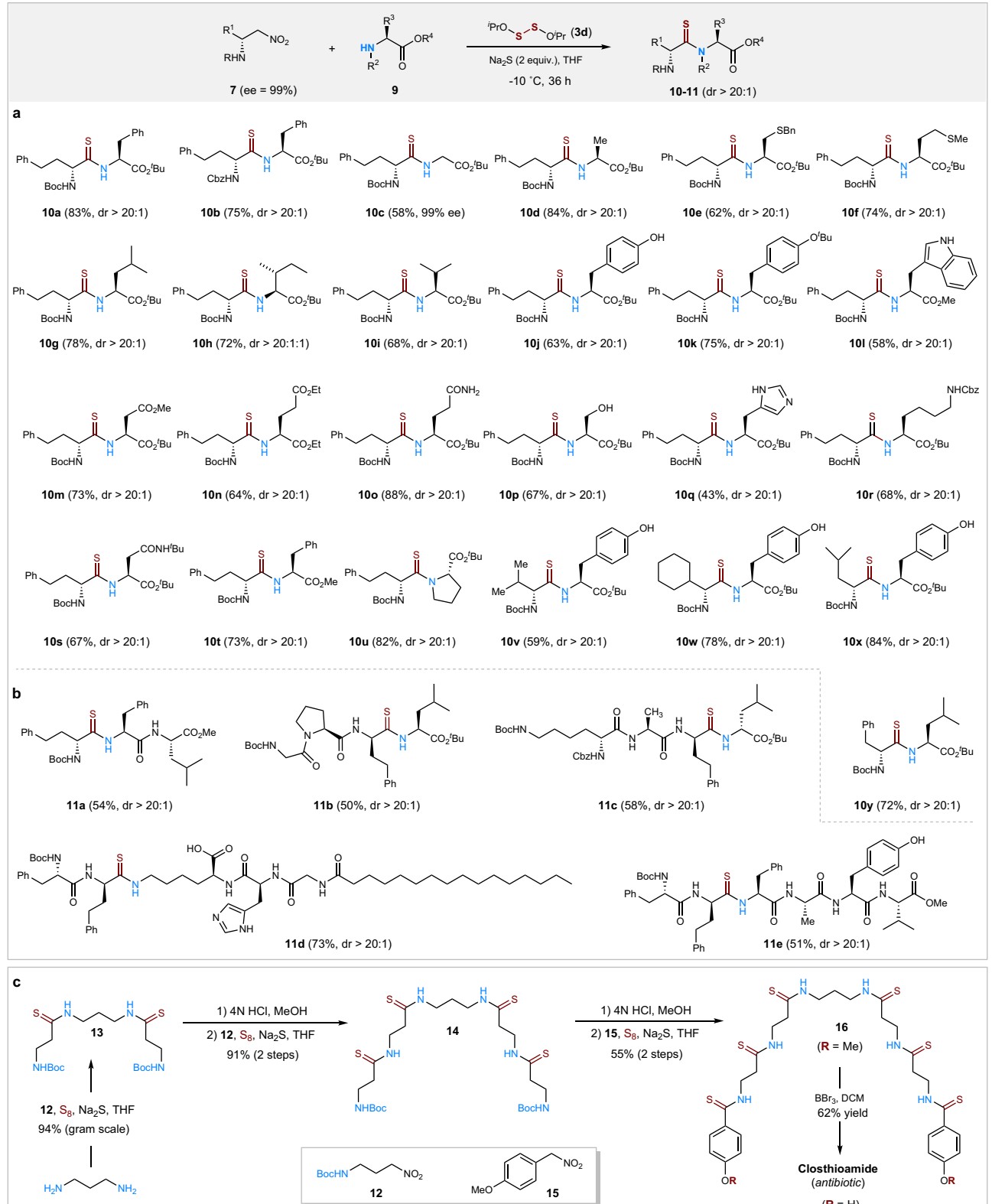

**Fig. 4 | General synthesis of *D/L*-thiopeptides from unnatural nitroalkane (*D*) precursors and *L*-amino acids.** Reaction conditions: **10a** (0.1 mmol), **11a** (0.2 mmol), Na$_2$S (0.2 mmol), **3d** (0.2 mmol) in THF (1 mL), stirred at −10 °C for 36 h.

Isolated yields are given in brackets. **a** Scope of thiopeptide bond formation. **b** Example of thioamide-substituted peptides. **c** Total synthesis of closthioamide.

trapped with amine to generate the thioamide product **4** (Fig. 5a, Path C). To interrogate this mechanistic possibility (Fig. 5d), the nitroalkane **22** was mixed with Na$_2$S and S$_8$ in DMSO-d$_6$ for NMR and HRMS studies (Reactions (5)). Here, we observed the conversion of nitroalkane **22** to

the dithiocarboxylate **23**. In addition, the nitroalkane **24** bearing a free γ-alcohol group was reacted with S$_8$ and Na$_2$S, either in the presence or absence of external nucleophiles (cf. Reactions (6) and (7)). Both cases gave the thiolactone **25**. These experiments imply the intermediacy of

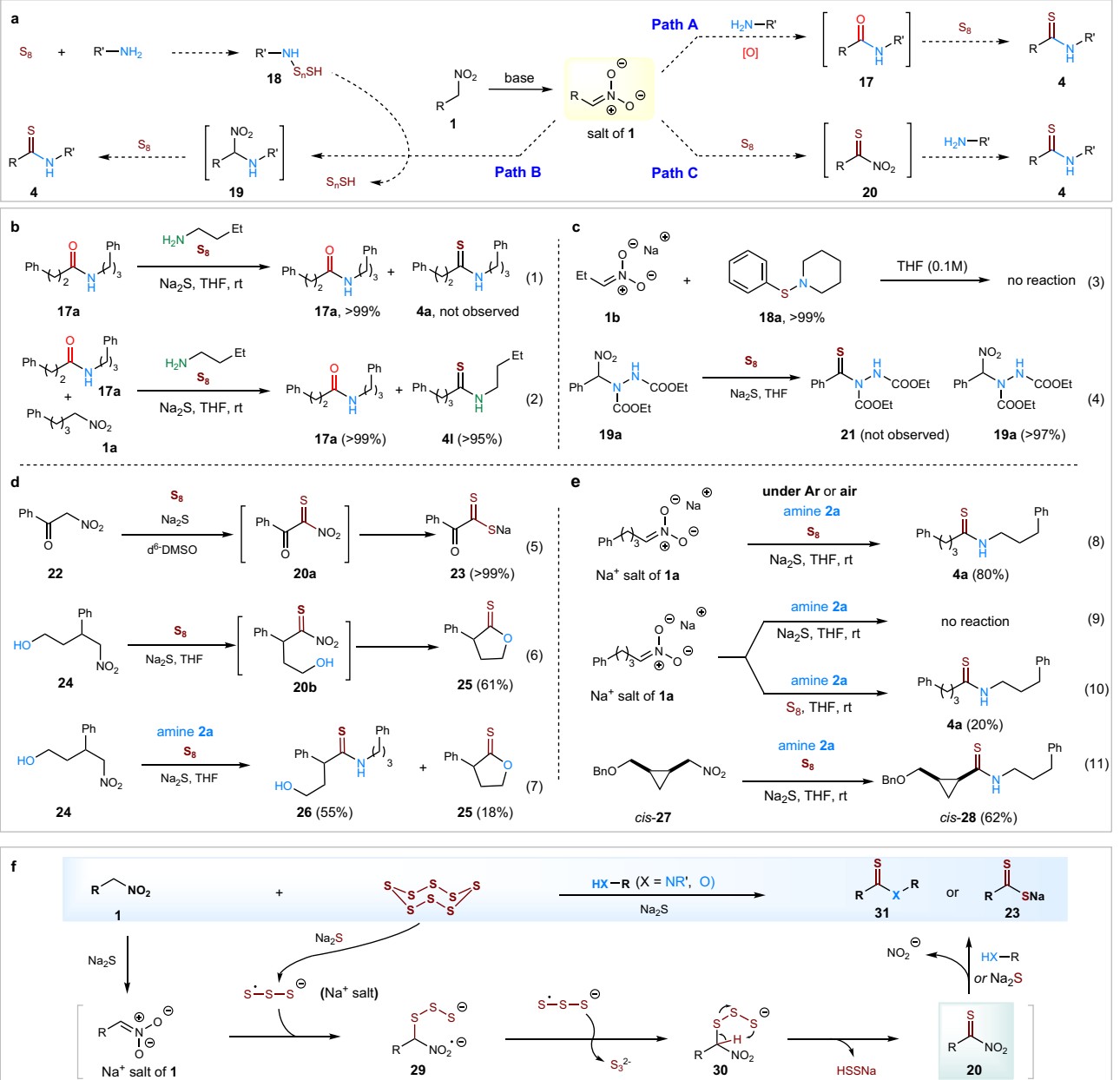

**Fig. 5 | Experimental studies to interrogate plausible reaction pathways and intermediates to form thioamides under S$_8$/Na$_2$S-based conditions. a** Possible mechanistic pathways for the formation of thioamides from primary nitroalkanes and amines. **b** Control reactions to interrogate Path A. **c** Control reactions to interrogate Path B. **d** Control reactions to trap proposed thioacyl intermediate **20** in Path C. **e** Control reaction for the formation of thioacyl intermediate **20**. **f** Proposed reaction pathway in the conversion of primary nitroalkanes to thioamides via thioacylating species **20**.

a reactive thioacyl species like **20**, which NO$_2$ leaving group as supported by ion chromatographic analysis on crude reaction mixtures, for example, a 91% yield of nitrite salts was formed between nitroalkane **1a** and amine **2a** under standard reaction conditions (Supplementary Fig. 6).

Having gained experimental evidence to support a putative thioacyl intermediate **20**, further control reactions were carried out to shed light on plausible intermediates prior to thiocarbonyl generation (Fig. 5e). First, Na$_2$S is considered to react with the starting nitroalkane to generate the α-carbanion/aci-nitronate sodium salt of **1a** in high concentrations. We therefore used the pre-formed Na$^+$ salt of **1a** to test its reaction with the amine **2a** in the presence of Na$_2$S/S$_8$ either under air or under Ar. Both cases gave the desired thioamide **4a** in 80% yield (see Reaction (8)). Second, the pre-formed Na$^+$ salt of **1a** when treated with amine **2a** and Na$_2$S in the absence of S$_8$ did not proceed at all. In

the absence of Na$_2$S, treatment of the pre-formed Na$^+$ salt of **1a** with amine and S$_8$ formed the thioamide very slowly (see Reaction (9) and Reaction (10)). These results indicate that both S$_8$ and Na$_2$S are required for a productive thioamidation process. Third, the β-cyclopropyl nitroalkane **27** was prepared as its pure *cis*-isomer for radical clock experiments to determine whether carbon-centered radicals form α to the nitro group[45–47]. In the event, when *cis*-**27** was reacted with the amine **2a** in the presence of Na$_2$S and S$_8$, it generated the *cis*-thioamide **28** exclusively in moderate yield (Reaction (11)), thereby providing evidence that α-carbon radicals do not form (Supplementary Fig. 8).

Based on the above control experiments, a general mechanism for this mild and direct thioamidation reaction is proposed in Fig. 5f. It is known that elemental sulfur can be activated by Na$_2$S to produce S$_3$ radical anions[48,49], which are expected to be the key reactive species in

the thioamidation process. Indeed, further control reactions and EPR experiments (Supplementary Fig. 10) suggested the temperature-dependent formation of $S_3$ radical anions when using either $S_8$ or the disulfide **3d**, thus indicating the reaction with **3d**/$Na_2S$ systems follows a similar mechanism to that given in Fig. 5f (cf. Supplementary Table 5, Supplementary Fig. 11). Thus, the sodium salt of **1**, generated in situ by deprotonation with $Na_2S$, couples with a $S_3$ radical anion at the α-carbanion position to afford a radical dianion salt **29**, which is oxidized by another $S_3$ radical anion to generate the organotrisulfide **30**[50–53]. Subsequent β-elimination of intermediate **30** then generates the thioacylating species **20** that can be captured by a suitable nucleophile to afford the product **31** or **23**. Although beyond the scope of this initial disclosure, our current studies do not discount other pathways to the formation of putative α-thio nitroalkanes like **30** and thioacyl species like **20**, including UmAS-based rationales.

In conclusion, the reaction method described herein is the general method to efficiently access thioamides and thiopeptides from readily available nitroalkanes and amines. The method is straightforward in operation, chemoselective in functional group tolerance, stereochemically robust to potentially epimerizable substrates and products, and avoids extensive protection and deprotection procedures. In addition, the use of commercial or readily synthesized chiral nitroalkanes as masked thioacylating agents establishes a practical alternative to the longstanding reliance of carboxylic acid feedstocks in thiopeptide chemistry. A wide range of chiral primary nitroalkanes can be readily prepared in enantiopure form through reliable asymmetric methods, which renders this methodology relatively practical for complex systems. The use of nitroalkanes as formal equivalents to activated thiocarbonyl groups now provides a general means to make thio-based value-added compounds and targeted chemical libraries, even beyond peptide chemistry. The application of this current method in the solid phase synthesis of thioamide-containing peptides is ongoing in our group and will be published in due course.

## Methods

### General thioamide coupling procedure

With no special precautions from air or water, the nitro compound **1** (0.2 mmol) is added to a 10 mL reaction tube, followed by the sequential addition of THF (2 mL), $S_8$ (2.0 equiv.), $Na_2S$ (2.0 equiv.) and the amine **2** (2.0 equiv.). The reaction is monitored by TLC until the nitroalkane appears consumed, typically after 24 h, after which the reaction is quenched with *sat.* $NH_4Cl$, extracted with ethyl acetate, and the organic phases collected and dried over anhydrous $Na_2SO_4$. After filtration, the solution is concentrated under reduced pressure and the crude residue purified by silica-gel flash-column chromatography.

### General thiopeptide coupling procedure

The chiral β-amino nitro compound **7** (0.1 mmol) is added to a 10 mL reaction tube, followed by the addition of THF (1 mL) and cooled down to −10 °C, after which $Na_2S$ (2.0 equiv., 0.2 mmol), **3d** (2.0 equiv.) and the amine **9** (2.0 equiv.) are added with stirring. The reaction is monitored by TLC until complete, typically within 36 h, quenched with *sat.* $NH_4Cl$, and the crude product extracted with ethyl acetate. After the organic phases are collected, dried over anhydrous $Na_2SO_4$ and filtered, the solution is concentrated under reduced pressure and the crude product purified by silica-gel flash-column chromatography.

## Data availability

All experimental procedures, characterization data, NMR spectra are available in the supplementary materials.

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

## Acknowledgements

Generous support by the Xi'an Jiaotong University is acknowledged. We are thankful to the National Natural Science Foundation of China (22101223) and the Start grant of Xi'an Jiaotong University. We thank Miss Lu Bai, Miss Chao Feng and Miss Na Li at the Instrument Analysis Center of Xi'an Jiaotong University for their assistance with HRMS, NMR and ions chromatography analysis. We also thank Pengfei Li from XJTU, Jin Xie from Nanjing University, Junfeng Zhao from GDMU and Xuefeng Jiang from ECNU for helpful discussions, as well as the University of Lincoln (UK) for support.

## Author contributions

J.L. conceived the idea and supervised the whole project. X.N.W. performed all experiments and analyzed the results regarding various substrates. X.N.W., J.L., S.L.X., and M.J.L. co-wrote the paper. Y.H.T. and W.X.H. contributed to discussion and revision of the paper. All authors approved the final version of the paper for submission.

## Competing interests

The authors declare no competing interests.
