## [Peer Review File · Nature Communications]

REVIEWER COMMENTS

Reviewer #1 (Remarks to the Author):

The manuscript by Li and co-workers reports on the development of a thioacylation methodology utilizing the reaction of nitroalkanes with amines directly in the presence of elemental sulfur and sodium sulfide to furnish thioamides. The substrate scope is comprehensive and the authors have clearly demonstrated both the general synthetic utility of the method and perhaps, more importantly, the generality and stereoselectivity of thiopeptide products. Mechanistic studies were carried out that support a putative thioacyl intermediate as a key component of the overall process and a reasonable mechanistic pathway involving S₃ radical anion is presented. Overall the study is well conducted and clearly supported by the ESI. I comment the authors on an excellent body of work and I recommend publication in Nature Communications.

Reviewer #2 (Remarks to the Author):

The authors describe a novel method for the preparation of thioamides that does not rely on thionation of oxoamides. Instead, this method draws inspiration from the work of Johnson on the use of halonitroalkanes in the preparation of amides, and the corresponding author's previous work with Hiyashi on the oxidative coupling of nitroalkanes and amines to generate amides.

The method is demonstrated to have wide substrate scope and under optimized conditions proceeds with no detectable loss of stereochemical integrity. Coupling of amino acids to generate dipeptide thioamides proceeds in good yield. One example of a peptide-peptide coupling is highlighted to generate a hexapeptide thioamide in reasonable yield. Polythiomides are also prepared in excellent yield.

This method will be of significant interest to medicinal and peptide chemists as an alternative route for the assembly of biologically active thioamides, including peptide thioamides.

One query for the authors regards solvent compatibility. All reactions are performed in THF (with some mechanistic studies undertaken in d₆-DMSO). Have other solvents been tested? Some substrates may not be soluble in THF so demonstration of solvent compatibility would be a valuable addition.

One aspect that is not fully explained is the superior performance of disulfide 3d in the optimized low temperature conditions. Were other sulfur sources tested in the coupling of chiral amino nitroalkanes (Fig. 3)? Can the authors explain why reactions with disulfide 3d proceed with higher yield than S8 at low temperature? Further, mechanistic studies were only performed with S8 as the sulfur source, not 3d, despite 3d being the sulfur source of choice for chiral substrates. The reaction with 3d presumably does not proceed through the S3 radical anion as postulated in the mechanism of the reaction with S8. The authors should therefore propose, with evidence, a mechanism for the reaction employing 3d.

Lastly, the mechanistic studies demonstrate formation of a dithiocarboxylate, yet the proposed mechanism suggests direct acyl substitution of the nitro thioacyl intermediate, without invoking formation of a dithiocarboxylate. Can the authors explain this conclusion?

Please find below point-by-point responses to the reviewers' comments.

Reviewer #1:

The manuscript by Li and co-workers reports on the development of a thioacylation methodology utilizing the reaction of nitroalkanes with amines directly in the presence of elemental sulfur and sodium sulfide to furnish thioamides. The substrate scope is comprehensive and the authors have clearly demonstrated both the general synthetic utility of the method and perhaps, more importantly, the generality and stereoselectivity of thiopeptide products. Mechanistic studies were carried out that support a putative thioacyl intermediate as a key component of the overall process and a reasonable mechanistic pathway involving S3 radical anion is presented. Overall the study is well conducted and clearly supported by the ESI. I commend the authors on an excellent body of work, and I recommend publication in Nature Communications.

→ We thank the reviewer for highlighting and pointing out the values of this key piece of work, which is proving to produce even more interesting results (unpublished work).

Reviewer #2:

The authors describe a novel method for the preparation of thioamides that does not rely on thionation of oxoamides. Instead, this method draws inspiration from the work of Johnson on the use of halonitroalkanes in the preparation of amides, and the corresponding author's previous work with Hiyashi on the oxidative coupling of nitroalkanes and amines to generate amides.

The method is demonstrated to have wide substrate scope and under optimized conditions proceeds with no detectable loss of stereochemical integrity. Coupling of amino acids to generate dipeptide thioamides proceeds in good yield. One example of a peptide-peptide coupling is highlighted to generate a hexapeptide thioamide in reasonable yield. Polythaiomides are also prepared in excellent yield. This method will be of significant interest to medicinal and peptide chemists as an alternative route for the assembly of biologically active thioamides, including peptide thioamides.

→ We highly appreciate the reviewer for pointing out the values of our current piece of work.

One query for the authors regards solvent compatibility. All reactions are performed in THF (with some mechanistic studies undertaken in d6-DMSO). Have other solvents been tested? Some substrates may not be soluble in THF so demonstration of solvent

compatibility would be a valuable addition.

→ It is indeed a very important suggestion. We tried polar solvents like DMF and THF mixtures, and all the solvents gave similar result for the thiopeptide synthesis. Please see the new data below.

entry	solvent	Yield (%)
1	THF	50
2	DMF	53
3	THF: DMF=1:1	51

We therefore noted this possibility of solvent choice in the supporting information (Supplementary Table 3) and in the main text as “*In practice, 11b could also be prepared in similar yields in DMF or DMF/THF mixtures, which is especially useful for substrates that are poorly soluble in THF (Supplementary Table 3)*”.

One aspect that is not fully explained is the superior performance of disulfide **3d** in the optimized low temperature conditions. Were other sulfur sources tested in the coupling of chiral amino nitroalkanes (Fig. 3)?

→ It is very good suggestion; we tried all the sulfur sources **3a**-to-**3d** and **S₈**; only **3d** gave the highest yield for thioamide **8a** formation at -10°C. Please see the expanded new Supplementary Table 2 in supporting information for more details. Based on the results, the disulfide **3d**/Na₂S system clearly confers a higher reactivity at -10°C (also see next response for additional mechanistic reasoning).

Can the authors explain why reactions with disulfide **3d** proceed with higher yield than **S₈** at low temperature? Further, mechanistic studies were only performed with **S₈** as the sulfur source, not **3d**, despite **3d** being the sulfur source of choice for chiral substrates.

The reaction with **3d** presumably does not proceed through the S₃ radical anion as postulated in the mechanism of the reaction with S₈. The authors should therefore propose, with evidence, a mechanism for the reaction employing **3d**.

→ The comments are valuable and helpful for mechanistic insights. It has been reported that elemental sulfur is activated with bases such as Na₂S and NaO^tBu to produce S₃ radical anions, which react with 1,3-diyne **S1** to afford thiophene **S2** (eq 1). (Ref. 48: *Org. Lett.* **2014**, *16*, 6156.)

Thus, we used this reaction (1) to detect the formation of sulfur-based radical anions, the most stable form being in an S₃ state. The results are summarized in Supplementary Table 4. When S₈ was used as sulfur source, the **S2** was obtained in moderate yield when NaO^tBu or Na₂S used as base. When **3d** was used as sulfur source, Na₂S as base, the **S2** was obtained in 70% yield at room temperature. When the temperature was decreased to -10 °C, and Na₂S used as base, **3d** gave a much better yield of thiophene **S2** than S₈. Taking all the results into account, both **3d** and S₈ can be activated with Na₂S and generate sulfur-based radical anions at room temperature. However, when the reaction is performed at low temperature, the formation of sulfur-based radical anions was lowered.

entry	sulfur source	base	temperature	NMR yield
1	S ₈	NaOBu ^t	rt	69%
2	S ₈	Na ₂ S	rt	68%
3	S ₈	Na ₂ S	-10 °C	5%
4	ⁱ PrO-S-S-O ⁱ Pr (3d)	Na ₂ S	rt	70%
5	ⁱ PrO-S-S-O ⁱ Pr (3d)	Na ₂ S	-10 °C	20

^aUnless noted otherwise, reactions were carried out with 0.1 mmol of **S1**, 0.2 mmol of base, and 0.2 mmol S source in 1 ml of DMF.

In addition, we performed the electron paramagnetic resonance (EPR) experiments below. This demonstrates a single EPR signal assignable to the trisulfur radical anion

observed in the DMF solution of Na₂S with **S₈** and **3d**, respectively, at room temperature. (Ref. 48: *Org. Lett.* **2014**, 16, 6156.)

Supplementary Fig.10 EPR experiment to detect the formation of S₃ radical anion.

We have now added this information to the new manuscript: “Indeed, further control reactions and EPR experiments (Supplementary Fig. 10) suggested the temperature-dependent formation of S₃ radical anions when using either **S₈** or the disulfide **3d**, thus indicating the reaction with **3d/Na₂S** systems follows a similar mechanism to that given in Fig.5f (cf. Supplementary Table 4, Supplementary Fig. 11).” and the new data has been added to the supporting information.

Based on the evidence above, we suggest a mechanism below, which we have added to the Supplementary Fig. 11. Thus, both S₈ and RSSR disulfides like **3d** (with R acting as leaving groups) are suggested to form S₃ radical anions according to the above evidence and correlation to references 48/49.

Lastly, the mechanistic studies demonstrate formation of a dithiocarboxylate, yet the proposed mechanism suggests direct acyl substitution of the nitro thioacyl intermediate, without invoking formation of a dithiocarboxylate. Can the authors explain this conclusion?

→ The dithiocarboxylate is actually a plausible side-product from the nitro thioacyl

intermediate (not an intermediate to the final product). To make this clearer, we have drawn the reaction mechanism to both a dithiocarboxylate **23** and the thioacyl adducts **31** in Fig. 5 via thioacyl intermediate **20**.

REVIEWERS' COMMENTS

Reviewer #2 (Remarks to the Author):

The authors have added additional experiments to address the concerns raised by the reviewers, including a demonstration of the reaction in a variety of solvents, comparison of further sulfur sources in the optimized conditions, and additional mechanistic investigations. Further, EPR analysis was conducted which shows the presence of the S3 radical anion. These additions significantly strengthen the conclusions drawn in the manuscript. The revised manuscript is suitable for publication.